# Estimating the Global Spread of Epidemic Human Monkeypox with Bayesian Directed Acyclic Graphic Model

**DOI:** 10.3390/vaccines11020468

**Published:** 2023-02-17

**Authors:** Ling-Chun Liao, Chen-Yang Hsu, Hsiu-Hsi Chen, Chao-Chih Lai

**Affiliations:** 1Institute of Epidemiology and Preventive Medicine, College of Public Health, National Taiwan University, Taipei 10055, Taiwan; 2Daichung Hospital, Miaoli County 35044, Taiwan; 3Master of Public Health Program, National Taiwan University, Taipei 10055, Taiwan; 4Emergency Department of Taipei City Hospital, Ren-Ai Branch, Taipei 10629, Taiwan

**Keywords:** monkeypox, Bayesian SIR, directed acyclic graphic method, reproductive number

## Abstract

A “Public Health Emergency of International Concern (PHEIC)” monkeypox outbreak was declared by the World Health Organization on 23 June 2022. More than 16,000 monkeypox cases were reported in more than 75 countries across six regions as of July 25. The Bayesian SIR (Susceptible–Infected–Recovered) model with the directed acyclic graphic method was used to estimate the basic/effective reproductive number (R_0_/R_e_) and to assess the epidemic spread of monkeypox across the globe. The maximum estimated R_0_/R_e_ was 1.16 (1.15–1.17), 1.20 (1.20–1.20), 1.34 (1.34–1.35), 1.33 (1.33–1.33) and 2.52 (2.41–2.66) in the United States, Spain, Brazil, the United Kingdom and the Democratic Republic of the Congo, respectively. The values of R_0_/R_e_ were below 1 after August 2022. The estimated infectious time before isolation ranged from 2.05 to 2.74 days. The PHEIC of the global spreading of human monkeypox has been contained so as to avoid a pandemic in the light of the reasoning-based epidemic model assessment.

## 1. Introduction

Monkeypox is a rare zoonotic disease, and the virus was initially discovered in 1958 [1]. The first human case of monkeypox, which is caused by the monkeypox virus, was diagnosed in 1970 [2]. Monkeypox mainly occurs in forested rural areas in Central and Western Africa. Most cases were acquired with the Central African clade and the others with the West African clade from 1970 to 2019 [3]. Since the cessation of smallpox vaccination, the incidence of reported monkeypox cases, the median age of patients, and the number of outbreaks are rising. In Africa, about 72.5% of monkeypox cases were suspected to be via animal-to-human transmission, and 27.5% cases were by a human source in the 1980s [4]. However, 78% of monkeypox cases were infected by a human source in the 1990s [5]. However, most cases outside of Africa were suspected to be via animal-to-human transmission [3]. Therefore, human-to-human transmission was inefficient because sporadic outbreaks and travel-associated cases outside Africa had limited secondary spread before April 2022. Unfortunately, more than 16,000 monkeypox cases were reported in more than 75 countries across six regions as of 25 July 2022 [6], and the majority of cases (98%) have been confirmed since early May 2022. In addition, the World Health Organization declared a “Public Health Emergency of International Concern (PHEIC)” monkeypox outbreak on 23 June 2022 [6]. Over 7000 monkeypox cases have been confirmed in the United States as of 5 August 2022 [7]. Over 90% of monkeypox cases have occurred mainly among men through sexual contexts [8,9]. Several reasons, including waning smallpox immunity, relaxing COVID-19 containment measures, and sexual activities after large gatherings, were linked to the global outbreak of monkeypox virus infection [8]. Avoiding human contact through good hygiene [10], early detection [9], and reducing the possibility of transmission by pre-exposure and post-exposure vaccination and treatment [11] are the major containment measures to prevent the outbreak of monkeypox. Thus, the surveillance of the spread of monkeypox is important to know the effectiveness of these containment measures.

The basic/effective reproductive number (R_0_/R_e_, the expected number of secondary infections) is a useful indicator for the spread of monkeypox. The aim of this study is to estimate the R_0_/R_e_ of monkeypox by the Bayesian SIR (Susceptible–Infected–Recovered) model to elucidate the spread of monkeypox. It is indispensable to determine whether or not monkeypox will continually spread through a population.

## 2. Materials and Methods

Data on the monkeypox outbreaks were retrieved from the Global Health team for estimating R_0_/R_e_. The details of data retrieval have been described in a previous report [12,13]. Data on the United States, Spain, Brazil, the United Kingdom and the Democratic Republic of the Congo worldwide were retrieved up to 23 September 2022 (Appendix A). Since there were limited established travel links to endemic areas in human-to-human transmission [8], we analyzed data with emphasis on non-endemic countries.

The initial investigation employed the deterministic SIR model as described by Kermack and McKendrick [14], and subsequent advancements were made through the incorporation of a Bayesian perspective. Both the deterministic SIR model and the Bayesian SIR model were used to estimate the R_0_/R_e_. Important assumptions in our deterministic SIR model and the Bayesian SIR model include the following: (1) all cases were infected by human-to-human transmission, and animal-to-human transmission was neglected; (2) the number of predicted cases was only detectable cases in this model; (3) we assumed similar transmission probability in the same country or globally; (4) a homogeneous random mixing population was assumed; (5) patients moved from I state to R state when they were isolated or recovered. Due to limited information on infectious periods before isolation or recovery, the infectious period of monkeypox was assumed to be the incubation period. The additional assumptions our deterministic SIR model also follows are (1) a fixed infectious period and that (2) the duration of these compartments are exponentially distributed. The minimal vaccine coverage considering the effectiveness against monkeypox is often derived by Vc = Ic/E = [1 − (1/Ro)]/E [15]), where Ic denoted the herd immunity threshold and E denoted the level of vaccine effectiveness. Following this rationale, we estimated the herd immunity threshold (Vc) using our maximum estimated R_0_/R_e_ to allow for the cross-protection afforded by smallpox vaccination.

### Bayesian SIR Model

The Bayesian ordinary differential equation (ODE) SIR model, including S (susceptible population), I (infectious persons with monkeypox), and R (recovered persons from infectious status or infectious persons being isolated) states, was used to estimate the basic reproductive number and to predict the global propagation of the monkeypox outbreak making use of the observed number of cases with the consideration of uncertainty [16]. Let *s*(t), *i*(t), and *r*(t) denote the numbers of susceptible, infectives, and recovered/isolated at time t. It can be depicted by using the ordinary differential equations [14] as follows:(1)$$\frac{ds\left(t\right)}{dt}=-\beta \;\cdot \;s\left(t\right)\;\cdot \;i\left(t\right)$$
(2)$$\frac{di\left(t\right)}{dt}=\beta \;\cdot \;s\left(t\right)\;\cdot \;i\left(t\right)-\alpha \;\cdot \;i\left(t\right)$$
(3)$$\frac{dr\left(t\right)}{dt}=\alpha \;\cdot \;i\left(t\right)$$

Note that as the population of susceptible is fixed, the following equation always holds: *s*(t) + *i*(t) + *r*(t) = *N*. Using a Bayesian framework, the parameters of transmission coefficient (*β*) and recovery rate (*α*) from the infectious status can be estimated by combining prior knowledge with current observation on the monkeypox epidemic. Two random variables, both the number of unknown infected status [*s*(t)] and the confirmed cases per day [*i*(t) + *r*(t)], follow two distributions of $Normal\left(\mu_{st},\;100\right)$ and $Poisson\left(\mu_{it}\right)$ [16], respectively. We set the initial value of s(0) as the total susceptible population, i(0) as the infectious case number, and r(0) with no recovery cases from the beginning.

The Bayesian directed acyclic graphic (DAG) model for assessing the spread of human monkeypox with a three-state (Susceptible–Infected–Recovered) model is illustrated in Figure 1. The transmission coefficient (*β*) and the recovery rate (*α*) were used to specify the Bayesian SIR model in accordance with the system of differential equations depicting the spread of monkeypox based on the SIR model (Equations (1)–(3)). Using the observed data on daily monkeypox cases after the first reported case throughout the duration of the epidemic (*i*_t_, *t* = 2,3,…,T, where T is the total number of days of observation, Figure 1) and the corresponding number of susceptible during the period (*s*_t_, *t* = 2,3,…,T), the mean count following the Poisson distribution, *μ*_It_ (*i*_t_~Poisson (*μ*_It_), and the mean value following the normal distribution, *μ*_St_ (*s*_t_~Normal (*μ*_St_, 100), were used to capture the expected number of monkeypox cases and the average number of susceptible population, respectively. These two parameters, *μ*_It_ and *μ*_St_, were in turn determined by both transmission coefficient (*β*) and recovery rate (*α*) by dint of the system of differential equations specified by (1) through (3). Based on the estimated result of transmission coefficient (*β*) and recovery rate (*α*), the reproductive number can further be derived by *β*/*α*.

On the specification of the Bayesian SIR model, a non-information prior (gamma (0.001, 0.001)) was used for the transmission coefficient parameter (*β*). Regarding the recovery rate parameter (*α*), an informative prior (gamma (26.7, 222.92)) was applied using the information on the inverse of the mean infectious period (8.5 days (95% CI: 6.6–10.9)) according to the result of the previous research [17].

Note that a deterministic Susceptible–Infected–Recovered (SIR) model was also used for the monkeypox outbreaks with similar nonlinear ordinary differential equations to the COVID-19 model, as indicated in Equations (1)–(3) above. The transmission coefficient (*β*) and R_0_ were obtained after fitting the observed cases with monkeypox by the simulation of this deterministic SIR model.

Monkeypox cases were predicted on a daily basis by using the Bayesian DAG-based posterior distribution, encrypted with parameters estimated from the previously reported data. In addition, it was also predicted by simulating the deterministic SIR model after fitting the total number of cases during the observed period. Two independent methods were used to evaluate the spread of monkeypox. The results obtained from the deterministic model were presented as auxiliary evidence for checking whether the result was consistent with the novel Bayesian DAG-based model.

The simulation of the deterministic SIR model was performed using the software MATLAB version 9.30.713579 (MathWorks, Natick, MA, USA), and the Bayesian SIR DAG model was analyzed using the SAS software (release 9.4).

## 3. Results

The results of the estimated parameters for monkeypox by using the deterministic SIR model in the global, United States, Spain, Brazil, the United Kingdom, and the Democratic Republic of the Congo are listed in Table 1. The estimated basic reproductive number (R_0_) or effective reproductive number (R_e_) of monkeypox ranged from 0.817 to 2.251 in the globe, given the fixed infectious period of 8.5 days. The estimated smallpox vaccine coverage required to contain the spread of monkeypox was highest in May 2022 at 56%. The maximum estimated R_0_/R_e_ values for monkeypox were 2.09 in the United States, 2.06 in Spain, 2.35 in Brazil, 2.43 in the United Kingdom and 2.80 in the Democratic Republic of the Congo, occurring between May and early June. The corresponding vaccine threshold for stopping monkeypox transmission in these countries ranged from 7% to 56%. The R_0_/R_e_ values decreased below 1 after August 2022.

The parameters estimated by the Bayesian SIR model from the monkeypox cases in the United States, Spain, Brazil, the United Kingdom, and the globe are listed in Table 2. The estimated basic reproductive number (R_0_) or effective reproductive number (R_e_) of monkeypox ranged from 1.00 (0.99–1.15) to 1.46 (1.37–1.51) on the global. The estimated required smallpox vaccine coverage for containing the spread of monkeypox was highest in May 2022 (37%). The maximum estimated R_0_/R_e_ was 1.20 (1.197–1.201), 1.19 (1.19–1.19), 1.34 (1.34–1.35), 1.33 (1.33–1.33), and 2.52 (2.41–2.66) in the United States, Spain, Brazil, the United Kingdom and the Democratic Republic of the Congo, respectively, which occurred between May and early June. The corresponding threshold of the vaccine was estimated as 31.4% and 56% by our different models (Table 1 and Table 2). The values of R_0_/R_e_ were below 1 after August 2022. Though the informative priors (8.5 days) were applied, the estimated infectious time before isolation ranged from 2.05 to 2.74 days (reciprocal *α*).

The cumulative monkeypox cases observed and those simulated by the deterministic SIR model for the entire world, as well as for the United States, Spain, Brazil, and the United Kingdom, are presented in Figure 2 and Appendix A. The cumulated observed cases and predicted cases by the Bayesian SIR model in different countries are shown in Figure 3 and Appendix A. We also predicted the cumulated cases on 30 September 2022, by parameters of the last period for the model validation, and we can realize the trend of monkeypox cases.

## 4. Discussion

The monkeypox outbreaks seemed to be controlled in the US, Spain, Brazil, the UK, the Democratic Republic of the Congo, and around the globe because the estimated R_0_/R_e_ gradually dropped to below 1. Though it has a long reciprocal period (up to 21 days) and a long contagious period, the estimated infectious time before isolation was short.

The basic reproduction number (R_0_) or the effective reproduction number (R_e_) is defined as the expected number of secondary infections per primary infection. It is often used to quantify the ability of an infectious disease to invade a population. R_0_ refers to the susceptible population without NPIs (nonpharmaceutical interventions) or vaccination, and R_e_ refers to the susceptible population with vaccination or some NPIs. R_0_/R_e_ above 1 indicates that the infectious disease is a potential epidemic. R_0_/R_e_ below 1 indicates that the infectious disease is potentially extinguished. The estimated R_0_/R_e_ was above 1 in the early stage of outbreaks, but it was below 1.5 in different countries in this study. In one previous study, the reproduction number of 1.29 was reported as of 22 July 2022 [18]. It is consistent with our results.

Monkeypox and smallpox belong to the orthopoxvirus family. Both of them can be transmitted by droplet exposure, close contact with infected skin lesions, or via contaminated materials. The R_0_ for smallpox was estimated between 3.5 and 6.0 in the previous study [19]. However, the spread of monkeypox is mainly through close or intimate contact with ill persons or animals. Hence, the R_0_ for monkeypox is lower than for smallpox.

The incidence of human monkeypox has risen since the 1970s, particularly in the Democratic Republic of the Congo [2]. The average age of those affected has risen from 4 years (the 1970s) to 21 years (2010–2019) [3]. The mortality rate stands at 8.7%, with a noticeable difference between different clades [3]. Outbreaks in countries outside Africa have been caused by importation cases. Activities or interactions with infected animals or individuals are factors that increase the risk of acquiring the disease. The discontinuation of smallpox vaccination, which provided protection against monkeypox, has led to a higher human-to-human transmission rate [20]. The occurrence of outbreaks outside Africa highlights the global significance of the disease burden for the enhanced surveillance of early detection.

The spread of monkeypox among persons in this global outbreak was well controlled by early diagnosis with isolation because the estimated infectious period before finding cases was below 3 days. Though the incubation period of monkeypox is usually from 6 to 13 days, it can range from 5 to 21 days [17]. In addition, high-risk populations get vaccinated to prevent the transmission of monkeypox.

Due to the eradication of smallpox, the global population after the year 1980 also has less immunity to monkeypox than the generation of those who have been vaccinated against smallpox. Mass vaccination against monkeypox is not recommended. To date, high-risk populations, including those who have an animal bite or scratch, travel to an endemic area, have close contact with infected people, health care workers caring for monkeypox patients, people with sexual behavior that puts them at risk of infection and immunocompromised people, are suggested to get vaccinated. In our estimation, reaching smallpox vaccine coverage (Vc) for high-risk people would be enough, and we also have people with immunity from before 1980.

There were three limitations in our study, including a homogeneous random mixing population assumption made for estimating the R_0_ of monkeypox, the heterogeneous surveillance systems across countries, and the only human-to-human transmission of monkeypox cases. In the original study design, only data from non-endemic countries with a lower proportion of animal-to-human transmission and a high proportion of confirmed monkeypox cases were analyzed because we were interested in modeling the epidemic process through human-to-human transmission. To test whether the estimate of basic reproductive number/effective reproductive number would be affected by including the endemic countries in Africa, we estimated the corresponding R_0_/R_e_ figure with the data from the Democratic Republic of the Congo, which had a higher proportion of suspected cases and a high probability of animal-to-human transmission [3] and may have led to a biased estimation of the basic reproductive number in this study. It is obvious that the pattern of the evolution of R_0_/R_e_ in such an endemic area is different from that in non-endemic areas, although there is not much difference in the maximum value between both types of estimates. The non-endemic process suggests the epidemic died out after the containment of human-to-human transmission, but the endemic process seems to remain rather stable. In addition, we would like to continue the study to the present day. However, the available data confirmed by the global team stopped being updated after 23 September 2022.

The proposed Bayesian SIR model is novel in that it not only gives up-to-date information on monkeypox transmission but also accounts for parameter uncertainty. The deterministic SIR model is used to describe the spread of infectious diseases where the output is a deterministically predicted value based on the inputs. On the other hand, the Bayesian SIR model is a probabilistic approach that not only adds uncertainty to the deterministic SIR model but also allows for updating the relevant parameters if more data are available. In this sense, the Bayesian SIR model generates posterior distributions for inference and prediction. Bayesian DAG was used to estimate the posterior distribution of parameters given observed data, which can be further used to make predictions about future outbreaks of monkeypox. Tailored by the clinical characteristics of monkeypox, the Bayesian DAG, in conjunction with ordinary differential equations, provides a flexible framework for modeling the spread of infectious disease. Furthermore, while the Bayesian DAG model has been extensively utilized in the field of machine learning, its application in conjunction with ordinary differential equations to model epidemic monkeypox outbreaks is novel and tailored for this purpose.

The R_0_/R_e_ estimated by the deterministic SIR model for the United Kingdom and Brazil in June was higher than the global R_0_ for all periods. Estimated R_0_/R_e_ by the deterministic SIR model was higher than the Bayesian SIR model possibly because of the fixed infectious period setting. The estimated R_0_/R_e_ derived from the deterministic SIR model was specified for observing the time change of parameters in each epoch.

We estimated the parameters based on the Bayesian SIR model for the uncertainty to minimize these biases, and the results were consistent in different countries. The estimated results were similar to the simulation data by the deterministic SIR model (Table 2, Figure 3).

## 5. Conclusions

In conclusion, the global spread of human monkeypox is not a threat because the R_0_/R_e_ is below one according to the Bayesian reasoning epidemic model assessment. Therefore, the epidemic has currently been contained and will not result in a pandemic because monkeypox is less contagious than smallpox, with available effective vaccines. As time goes on, continuous surveillance of the outbreak will still be needed.

## Figures and Tables

**Figure 1 vaccines-11-00468-f001:**
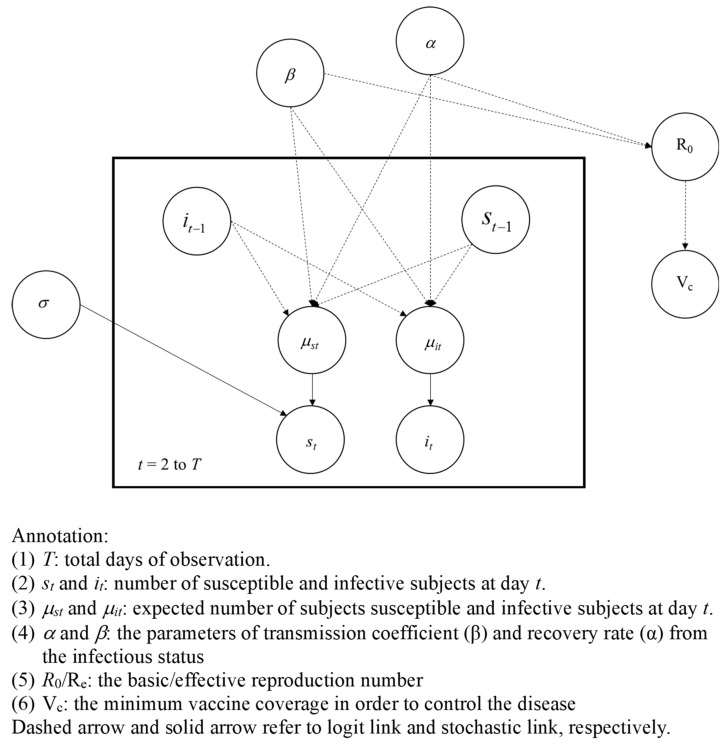
The directed acyclic graph model for estimating the spread of human monkeypox.

**Figure 2 vaccines-11-00468-f002:**
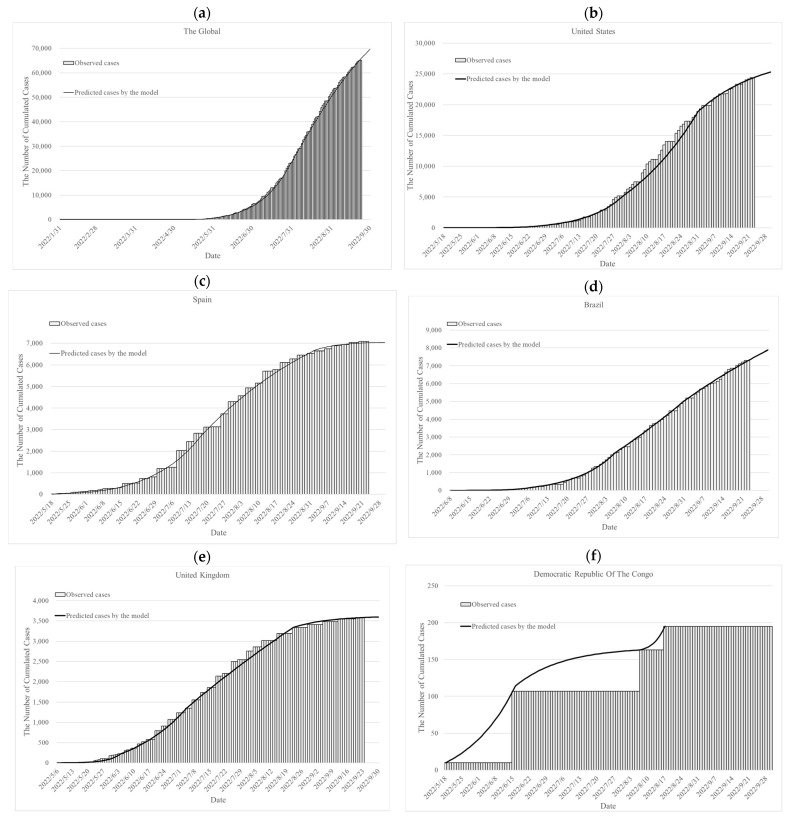
The observed cumulative monkeypox cases and predicted cumulated monkeypox cases on the global and in the different countries by the deterministic SIR model: (**a**) globally, (**b**) in the United States, (**c**) in Spain, (**d**) in Brazil, (**e**) in the United Kingdom and (**f**) in the Democratic Republic of the Congo.

**Figure 3 vaccines-11-00468-f003:**
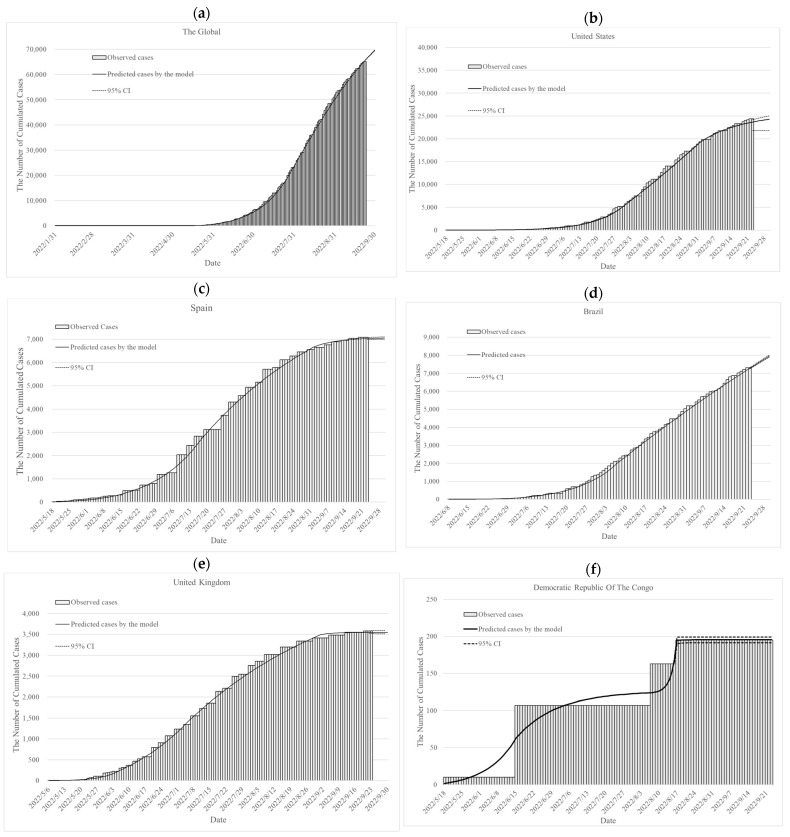
The observed cumulative monkeypox cases and predicted cumulated monkeypox cases on the global and in different countries by the Bayesian SIR model: (**a**) globally, (**b**) in the United States, (**c**) in Spain, (**d**) in Brazil, (**e**) in the United Kingdom and (**f**) in the Democratic Republic of the Congo.

**Table 1 vaccines-11-00468-t001:** Results of estimated parameters from monkeypox outbreaks by using the deterministic SIR model.

Country/Areas	The Periods		The Deterministic SIR Model	
		*β*	R_0_/R_e_	Vc(%)
The Globe	31 January~30 April	0.132	1.119	11
	1 May~31 May	0.265	2.251	56
	1 June~30 June	0.180	1.532	35
	1 July~31 July	0.149	1.271	21
	1 August~31 August	0.136	1.153	13
	1 September~23 September	0.096	0.817	-
United States	18 May~30 June	0.245	2.086	52
	1 July~30 July	0.184	1.568	36
	1 August~31 August	0.145	1.229	19
	1 September~23 September	0.080	0.677	-
Spain	18 May~16 June	0.242	2.057	51
	17 June~16 July	0.15318	1.302	23
	17 July~31 August	0.11767	1.000	-
	1 September~23 September	0.0523	0.445	-
Brazil	8 June~7 July	0.276	2.346	57
	8 July~6 August	0.1882	1.600	38
	7 August~31 August	0.1271	1.080	7
	1 September~23 September	0.1077	0.915	
United Kingdom	8 June~7 July	0.286	2.432	59
	8 July~6 August	0.154	1.312	24
	7 August~31 August	0.113	0.964	
	1 September~23 September	0.055	0.470	
Democratic Republic of the Congo	8 May~5 June	0.118	1.407	29
	6 June~28 July	0.067	0.568	
	28 July~7 August	0.329	2.797	64

R_0_: basic reproductive number; R_e_: effective reproductive number; SIR model: Susceptible–Infected–Recovered model; Vc: The minimum vaccine coverage in order to control the disease.

**Table 2 vaccines-11-00468-t002:** Results of estimated parameters from monkeypox outbreaks by using the Bayesian SIR model.

Data	Total Cases		Estimated by the Bayesian SIR Model			
		The Periods	*β* (95% CI)	*α* (95% CI); Reciprocal *α* (95% CI)	R_0_/R_e_ (95% CI)	Vc (%) (95% CI)
The Globe	65,215	31 January~30 April	0.374 (0.368–0.384)	0.373 (0.373–0.374);	1.001 (0.986–1.150)	-
		1 May~31 May	0.545 (0.511–0.563)		1.459 (1.370–1.507)	31.4 (27.0–33.6)
		1 June~31 July	0.420 (0.419–0.421)		1.125 (1.123–1.127)	11.1 (10.9–11.3)
		1 August~23 September	0.383 (0.383–0.384)	2.678 (2.677–2.680)	1.027 (1.026–1.028)	2.6 (2.5–2.7)
United States	24,403	17 May~30 June	0.583 (0.581–0.589)	0.487 (0.485–0.490);	1.198 (1.197–1.201)	16.5 (16.4–16.7)
		1 July~30 July	0.567 (0.563–0.569)		1.164 (1.149–1.169)	14.1 (13.0–14.5)
		31 July~31 August	0.499 (0.493–0.516)		1.024 (1.016–1.052)	2.3 (1.6–4.9)
		1 September~23 September	0.407 (0.192–0.452)	2.054 (2.039–2.061)	0.837 (−0.391–0.929)	-
Spain	7083	18 May~16 June	0.480 (0.478–0.483)	0.403 (0.400–0.405);	1.191(1.189–1.194)	16.0 (15.9–16.2)
		17 June~16 July	0.454 (0.451–0.457)		1.126 (1.124–1.129)	11.2 (11.0–11.4)
		17 July~31 August	0.389 (0.386–0.391)		0.963 (0.961–0.966)	-
		1 September~23 September	0.298 (0.278–0.316)	2.479 (2.467–2.498)	0.739 (0.690–0.784)	-
Brazil	7300	8 June~7 July	0.489 (0.487–0.492)	0.365 (0.363–0.368);	1.341 (1.337–1.345)	25.4 (25.2–25.6)
		8 July~6 August	0.435 (0.433–0.439)		1.193 (1.189–1.197)	16.2 (15.9–14.4)
		7 August~23 September	0.362 (0.359–0.365)	2.740 (2.717–2.758)	0.992 (0.990–0.994)	-
United Kingdom	3585	8 June~7 July	0.542 (0.539–0.545)	0.408 (0.405–0.411);	1.330 (1.326–1.334)	24.8 (24.7–25.0)
		8 July~6 August	0.437 (0.433–0.440)		1.071 (1.067–1.074)	6.6 (6.3–6.9)
		7 August~31 August	0.395 (0.392–0.398)		0.969 (0.966–0.972)	-
		1 September~23 September	0.181 (0.115–0.280)	2.453 (2.436–2.472)	0.443 (0.284–0.687)	-
Democratic Republic of the Congo	195	8 May~5 June	0.482 (0.463–0.497)	0.397 (0.376–0.415)	1.216 (1.199–1.235)	17.8 (16.6–19.0)
		6 June~28 July	0.334 (0.313–0.358)		0.841 (0.816–0.864)	
		29 July~7 August	0.847 (0.794–0.903)		2.136 (2.011–2.284)	53.2 (56.2–78.3)
		After 8 August	0.005 (0.000–0.033)	2.524 (2.412–2.662)	0.013 (0.000–0.083)	

R_0_: basic reproductive number; R_e_: effective reproductive number; SIR model: Susceptible–Infected–Recovered model.; CI: credible interval; Vc: The herd immunity threshold.

## Data Availability

All data relevant to the study are included in the article.

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
