# Peer review of "Estimating the Global Spread of Epidemic Human Monkeypox with Bayesian Directed Acyclic Graphic Model"

_vaccines, 2023, doi:10.3390/vaccines11020468_

Round 1

Reviewer 1 Report

The article is devoted to applying the Bayesian SIR model using the method of directed acyclic graphics to estimate the basic/adequate reproductive number (R0 / Re) and the epidemic distribution of monkeypox in the world. The study's relevance is justified by the fact that on June 23, 2022, the World Health Organization announced an outbreak of a "Public Health Emergency of International Concern (PHEIC)". As of 25 July, more than 16,000 cases of monkeypox had been reported in over 75 countries in six regions. As a result of the study, the maximum estimated R0/Re was 1.16 (1.15–1.17), 1.20 (1.20–1.20), 1.34 (1.34–1.35), and 1 .33 (1.33–1.33) in the US, Spain, Brazil, and the UK, respectively. R0/Re values were closer to 1 after August 2022. The estimated infectious time to isolation ranged from 2.05 to 2.74 days.

Despite the satisfactory quality of the article, some shortcomings need to be corrected.

  1. It is recommended to include the Current research analysis section.
  2. The model described by the system of ordinary differential equations is the classic SIR model presented by Kermack and McKendrick.
  3. The model presented in Figure 1 should be described in detail in the Materials and Methods section.
  4. The novelty of the proposed approach should be highlighted. It is unclear what the difference is between the described models from the known ones.
  5. The data on Monkeypox morbidity and data sources should be described in detail.
  6. It is unclear how authors forecast the cases presented in Figure 2. The graphs look like regression models, not the Bayesian SIR model.
  7. The authors concluded that “human Monkeypox outside Africa is not a threat”. However, none of the African countries was evaluated in the research.
  8. The limitations of the described approach should be described.

In summarizing my comments, I do not think the given paper can be accepted in current form. 

Author Response

Thanks for the valuable suggestion. We followed the reviewer’s suggestion to revise our manuscript. In addition, we also revised our manuscript to correct the grammar and phrasing. Indicate the changes to our manuscript by using a red color font. We explained and revised it as follows:

Q 1: It is recommended to include the Current research analysis section.
A1: Thanks for the reviewer’s valuable suggestion. We added it as follows. [Page 4, Line 182]
“The incidence of human monkeypox has risen since the 1970s, particularly in the Democratic Republic of the Congo. [2] The average age of those affected has risen from 4 years (the 1070s) to 21 years (2010-2019). [3] The mortality rate stands at 8.7%, with a noticeable difference between different clades.[3] Outbreaks in countries outside Africa have been caused by importation cases. Activities or interactions with infected animals or individuals are factors that increase the risk of acquiring the disease. The discontinuation of smallpox vaccination, which provided protection against monkeypox, has led to a higher human-to-human transmission rate. [19] The occurrence of outbreaks outside Africa highlights the global significance of the disease burden for the enhanced surveillance of early detection.”

Q2: The model described by the system of ordinary differential equations is the classic SIR model presented by Kermack and McKendrick.
A2: Thanks for the reviewer’s suggestion. We revised it as follows, in the section on Materials and Methods. (Page 2, Line 62).
“The initial investigation employed the deterministic SIR model as described by Kermack and McKendrick [14], and subsequent advancements were made through the incorporation of a Bayesian perspective.”

Q 3: The model presented in Figure 1 should be described in detail in the Materials and Methods section.
A3: Thanks for the reviewer’s comments. We have already described it in the section on Materials and Methods. (Page 3, Line 97).
“The Bayesian directed acyclic graphic (DAG) model for estimating the spread of human monkeypox 3-state model is illustrated in Figure 1. The equation derived earlier description of the correlation between the transmission coefficient (β) and recovery rate (α) with the predicted number of susceptible and infective individuals at day t. These two components serve as the scale parameters for the normal distribution that describes the susceptible population (St) and the Poisson distribution that describes the infective population (it). The basic reproductive number (R0/Re) can be calculated by combining the transmission coefficient and recovery rate through a logical function.”

Q4: The novelty of the proposed approach should be highlighted. It is unclear what the difference is between the described models from the known ones.
A4: Thanks for the reviewer’s valuable suggestion. We revised it in the section of the discussion.(Page 4, Line 211), as follows.
“The deterministic SIR model is used to describe the spread of infectious diseases, where the output is a deterministically predicted value based on the inputs. On the other hand, the Bayesian SIR model is a probabilistic approach that not only adds uncertainty to the deterministic SIR model but also allows for updating the relevant parameters if more data are available. In this sense, the Bayesian SIR model generates posterior distributions for inference and prediction. Bayesian DAG was used to estimate the posterior distribution of parameters given observed data, which can be further used to make predictions about future outbreaks of Monkeypox. Therefore, the Bayesian SIR model gets over advantage of the deterministic SIR model is our novelty in tackling the uncertainty of parameters.”

Q5: The data on Monkeypox morbidity and data sources should be described in detail.
A5: Thanks for the reviewer’s comments. The current data of morbidity are based on endemic countries such as Democratic Republic of Congo, which were limited in non-endemic countries. We use data retrieved from the Global health team and retrieved till September 23, 2022. We have already revised it in the section of Materials and Methods. (Page 2, Line 56) as follows.
“Data on the Monkeypox outbreaks were retrieved from the Global Health team for estimating R0/Re. The details of data retrieval have been described in the previous report [12,13]. Data on the United States, Spain, Brazil, and the United Kingdom worldwide were retrieved till September 23, 2022 (Table 1). Since there were limited established travel links to endemic areas in human-to-human transmission [8], we analyzed data with emphasis on non-endemic countries.”

Q6: It is unclear how authors forecast the cases presented in Figure 2. The graphs look like regression models, not the Bayesian SIR model.
A6: Thankful for the reviewer’s comments. Figure 2 data was retrieved by our simulation of cumulated cases of monkeypox from the deterministic SIR model. We have already described it in detail in the section on the method. (Page 3, Line 134) as follows.
“The cumulative monkeypox cases observed and those simulated by the determin-istic SIR model for the entire world, as well as for the United States, Spain, Brazil, and the United Kingdom, are presented in Figure 2.”

Q7: The authors concluded that “human Monkeypox outside Africa is not a threat”. However, none of the African countries was evaluated in the research.
A7: Thanks for the reviewer’s comments. We did not evaluate any African countries, as Monkeypox was geographically limited to endemic countries in West and Central Africa. According to the WHO report based on non-endemic countries and limited established travel links to endemic areas since May 13, 2022, the conclusion was briefly drawn and can be only applied to the countries outside Africa. We have revised it in the section on the method. (Page 2, Line 59) as follow.
“Since there were limited established travel links to endemic areas in human-to-human transmission[8], we analyzed data with emphasis on non-endemic countries.”

Q8: The limitations of the described approach should be described.
A 8: Thanks for the reviewer’s suggestion. We have described it in the section of Discussion. (Page 4, Line 205) as follow.
“There were three limitations in our study including a homogeneous random mixing population assumption made for estimating the R0 of monkeypox, the hetero-geneous surveillance systems across countries, and only the human-to-human trans-mission of monkeypox cases outside Africa in this study. In addition, we would like to continue the study to the present day. However, the available data confirmed by the global team discontinued updating after September 23, 2022.”

Reviewer 2 Report

This manuscript brings results that estimate the basic/effective reproductive number of the epidemic spread of monkeypox in the world. The regions also evaluted were USA, Spain, Brazil and UK. This is an useful protocol in order to survey the spread and possible risk to turn into a pandemic disease. 

I just ask the authors if they intend to continue the study till the present days

Minor detail: correct capital letters in reference 8

Author Response

Q1: I just ask the authors if they intend to continue the study till the present days
A1: Thanks for the reviewer’s suggestion. We are thankful for the reviewer’s comments. We would like to continue the study to the present day. However, the available data confirmed by the global team discontinued updates after September 23, 2022. We added it in the section of the discussion as follows. (Page4, line 208)
“In addition, we would like to continue the study to the present day. However, the available data confirmed by the global team discontinued updating after September 23, 2022.”

Q2: Minor detail: correct capital letters in reference 8.
A2: Thanks for the reviewer’s comments. We revised it as follows.
“8. Thornhill, J.P.; Barkati, S.; Walmsley, S.; Rockstroh, J.; Antinori, A.; Harrison, L.B.; Palich, R.; Nori, A.; Reeves, I.; Habibi, M.S.; et al. Monkeypox Virus Infection in Humans across 16 Countries - AprilJune 2022. N Engl J Med 2022, 387, 679-691, doi:10.1056/NEJMoa2207323.” 

Reviewer 3 Report

Good study

Author Response

Q1: Good study
A1: Thanks for the reviewer’s comments.

Round 2

Reviewer 1 Report

The authors have not appropriately responded to the comments and recommendations provided by the reviewer. In my opinion, the paper can not be published because:

1. The proposed model does not have any novelty. It is a well-known model which is not changed according to the specifics of the monkeypox epidemic process spreading.

2. The results presented in the study are unclear. Forecasts presented in Figure 2 are not forecasts because the line which indicates the forecast starts from the beginning of the epidemic process spreading.

3. The design of the study is wrong. The authors make conclusions about countries outside Africa, not analyzing African countries.
